# Effect of Short-Term Aging on Asphalt Modified Using Microwave Activation Crumb Rubber

**DOI:** 10.3390/ma12071039

**Published:** 2019-03-29

**Authors:** Bo Li, Jianing Zhou, Zhihao Zhang, Xiaolong Yang, Yu Wu

**Affiliations:** 1National and Provincial Joint Engineering Laboratory of Road & Bridge Disaster Prevention and Control, Lanzhou Jiaotong University, Lanzhou 730070, China; zjn1409@163.com (J.Z.); tmgcxyyb@163.com (Z.Z.); glhyyfzx@gslq.com (X.Y.); gsgfjt@163.com (Y.W.); 2Research and Development Centre of Transport Industry of Technologies, Materials and Equipment of Highway Construction and Maintenance, Gansu Road and Bridge Group Co., Ltd., Lanzhou 730070, China; 3Key Laboratory of Highway Net Monitoring in Gansu Province, Gansu Hengda Road and Bridge Group Co., Ltd., Lanzhou 730070, China

**Keywords:** microwave activation, short-term aging, rheological property, repeated creep recovery

## Abstract

Effective approaches are required to be developed to solve the poor compatibility and thermal stability problems of crumb rubber-modified asphalt (CRMA). This study focuses on a method called microwave activation. However, seldom researches pay attention to the properties of MACRMA after aging. The objective of this study was to prepare microwave-activated CRMA (MACRMA) and investigate the performance of asphalt after aging. The samples were subjected to thin-film oven test (TFOT) at different times and temperatures. The effect of heat aging on the properties of MACRMA was evaluated by three indicator tests: viscosity, dynamic shear rheology test (DSR), and repeat creep recovery test (RCRT). The test results indicated that the MACRMA after two aging conditions had noticeably lower performance values (e.g., penetration, ductility) compared to unaged samples, and thus, the need to control temperature and time for mixing and construction was verified to be important. In addition, the *G**/sin δ and phase angle values were largely influenced by the TFOT aging temperature and time. The MACRMA’s ability to recover was improved after aging. Compared with the aging temperature, the aging time had a more significant effect on the deformation and recovery ability of MACRMA.

## 1. Introduction

Approximately 1.4 billion tires are produced every year for vehicles of industrialized and developing countries, which leads to more tires that need to be recycled or disposed [1,2,3,4]. Disposal of used and discarded tires have posed a significant environmental threat worldwide every year due to fire hazards and have become a habitat for insects/rodents. In recent years, many modern technologies have been invested in producing crumb tire from tire rubber while removing fibers, steel, and other contaminants [5,6,7,8]. One of the means for utilizing crumb rubber is as a modifier for producing rubberized asphalt, since crumb tires contains a variety of rubber polymers, predominantly natural rubber and styrene-butadiene rubber.

Currently, the technology of crumb rubber-modified asphalt (CRMA) has been widely used in the asphalt pavement industry [9,10,11,12,13,14,15]. The addition of crumb rubber is typically accomplished using the wet process, in which the crumb rubber is blended with the asphalt binder to produce a crumb rubber modified binder that is then mixed with the aggregate. In this wet process, asphalt binder modification is due to physical and compositional changes from interactions between rubber and asphalt, characterized by a swelling of the rubber particles in the asphalt binder due to absorption of some lighter fractions (aromatic oils) from asphalt. The swelling of these rubber particles and their absorption, in turn, results in a viscous gel. CRMA developed using the wet process has been proven by many research projects and field applications to be an effective method to increase the performance grade of the asphalt, improve the high temperature properties, decrease susceptibility to permanent deformation, and provide resistance against reflective cracking. In addition, using crumb rubber as a modifier instead of expensive polymer modifiers (SBS, SBR, etc.) can reduce the cost of the road asphalt [16,17,18,19].

However, CRM has poor compatibility with asphalt since the waste rubber powder is an inert material, which limits the partial performance of CRMA. The final performance of asphalt with CRM modified binder is greatly influenced by the physical and chemical properties of CRM. As crumb rubber molecules are vulcanized, the three-dimensional network is difficult to crack in the asphalt, even after blending at high temperatures for a long period of time. This leads to incompatibility between CRM and asphalt, and settlement of rubber particles at the bottom of the bulk asphalt phase, which further affects the performance of CRMA [20,21,22,23,24]. The literature shows that in order to minimize the problem of CRM particle sedimentation, the prepared CRM binder should be used within four hours. Many attempts have been made to improve the properties of CRM, such as regenerated desulphurization modifications and grafting to break the C–S or S–S bonds in the rubber chemical linking networks. However, these modification methods are done by adding CRM to rubber or plastic but not to asphalt. In addition, the modification process is complicated and difficult to control. In recent years, researchers have used microwaves to modify CRM, and then add the activated CRM to asphalt. Many studies have shown that microwave activation is similar to previous methods that break the internal S–S bond of the rubber powder and improve the surface activity, and the storage stability of the microwave activated CRM asphalt (MACRMA) and pavement performance is better [25,26,27].

Despite its significantly modified properties, MACRMA is vulnerable to aging, which is an inevitable process in its service life that can significantly deteriorate the asphalt binding properties. Asphalt has always been in adverse conditions due to excessive temperatures and long heating times in the actual production, transportation, and construction, which will inevitably cause short-term aging of the asphalt and weaken the function of the asphalt binder and mixture. Short-term aging mainly takes place during the manufacturing process of asphalt mixtures in which the aggregates are covered by a thin asphalt film at high temperatures. The thin asphalt film has large surface areas that are exposed to air, causing the oxidative aging of asphalt components and the volatilization of light fractions. It is noted that the performance of asphalt mixtures in pavement after short-term aging was determined universally [28,29].

Numerous laboratory aging methods have been proposed to simulate asphalt aging. Among these methods, the rolling thin-film oven test (RTFOT) is the most commonly applied as recommended by the Strategic Highway Research Program [30]. However, researchers have shown that the RTFOT is not suitable for CRM asphalt owing to its relatively low aging temperature of 163 °C. The aging of RTFOT relies on spreading the fluid asphalt into thin films. With a reduced asphalt film thickness, the exposure to oxygen is increased and thus the aging process is accelerated. The CRM is usually too stiff to flow at 163 °C inside the RTFOT aging bottle, and thus it cannot form thin films, resulting in insufficient aging. Many studies have indicated that the current 163 °C aging temperature should be elevated because it was established on field investigations of neat asphalt and is apparently lower than the CRM mixing temperature. As MACRMA brings notably enhanced engineering properties and higher viscosity, this results in increased mixing and compaction temperatures. Contractors in China have reported that CRM asphalt usually has a mixing temperature of 190 °C and a paving temperature of 170–185 °C, depending on the time spent in transport. RTFOT does not have a comparable aging temperature in the field and fails to form an asphalt film of a small uniform thickness for asphalt binders with different viscosities, and its application for MACRMA is questionable [31].

The objective of this study is to investigate the influence of different aging factors on the properties of MACRMA from the perspective of rheological properties, and gradually exploring the thermal oxygen aging law of MACRMA. The MACRMA was subjected to a thin film oven test (TFOT) for different times (0, 5, 15, and 20 h) and temperatures (150, 160, 170, 180, and 190 °C). A serious of lab tests including three indicator tests: viscosity, dynamic shear rheology (DSR) test, and repeated creep recovery test have been conducted.

## 2. Materials and Experiments

### 2.1. Materials

The neat asphalt binder used in this study was petroleum asphalt of penetration grade 90. Table 1 presents some detailed technical properties of neat petroleum asphalt of penetration grade 90. Each index was tested based on the Standard Test Method of Bitumen and Bituminous Mixture for Highway Engineering (JTG E20-2011) by the Chinese Ministry of Communications [32]. The 40-mesh crumb rubber powder produced by the normal temperature method was selected, and the physical and chemical properties of the test results are shown in Table 2. At the same time, considering the influence of the production method of the crumb rubber powder on the particle size, the crumb rubber powder is sieved through the 40-mesh filter prior to using.

### 2.2. Experimental Procedures

The waste tire rubber powder was placed in a constant-temperature oven at 60 °C for 30 min for drying and dehydration, and then 100 g of the rubber powder was placed in the microwave oven. In this experiment, the activated rubber powder was prepared by activating the rubber powder for 90 s at 800 W and then cooling to room temperature. The asphalt binder was heated to approximately 135 °C as measured with a thermometer and poured into a beaker. Then, the beaker was placed in a constant temperature magnetic heating stirrer to raise the temperature to 190 °C. Additionally, the activated rubber powder was slowly added at a stirring rate of 1500 r/min. After reacting with the asphalt for 60 min, the microwave rubber modified asphalt was obtained. The cigar tube test showed that the difference of softening point of the upper and lower ends of the tube was only 2.3 °C, which indicated that the modified asphalt had been uniformly mixed. Some of its performance indicators are shown in Table 3. 

To simulate the effect of short-term aging on the performance of MACRMA, the asphalt was aged by the thin film oven test (TFOT, Figure 1). This test uses a single variable method: (1), keeping the aging time at 5 h and changing the aging temperature (150, 160, 170, 180, and 190 °C); (2), keeping the aging temperature at 163 °C and changing the aging time (5, 10, 15, and 20 h).

The dynamic shear rheology test (DSR) was used to test the complex shear modulus *G**, phase angle (δ), and other rheological properties with respect to AASHTO T315. The temperature sweep, frequency sweep and repeated creep recovery test of MACRMA samples were carried out by AR1500ex dynamic shear rheometer. The temperature scanning range was 58–82 °C and the frequency was 10 rad/s. The frequency sweep range was 0.1~10 rad/s with a test temperature of 70 °C and 12% strain.

The RCRT test is an abbreviation for the repeated creep recovery test specified by AASHTO MP19-1. The test temperature was set to 60 °C, and the loading stress was 100 Pa and 300 Pa, respectively. In addition, the loading parallel plate diameter was 25 mm, and the test spacing was 1 mm. One creep recovery cycle included the 1 s loading mode and the 9 s unloading mode, which gave 50 time measurements in total. This loading mode can better simulate the intermittent characteristics of actual asphalt pavement load. In other words, the characteristics of higher elastic recovery of the modified asphalt could be considered by this test.

## 3. Experiment Results

### 3.1. Penetration/Softening Point/Ductility Tests

These three indicator tests are the most widely used methods for evaluating asphalt performance in China. The penetration test is the standard method for determining the consistency of asphalt. The penetration degree indicates the consistency of the asphalt. The larger the index, the lower the viscosity of the asphalt.

The penetration values in terms of increased aging temperature and time are shown in Figure 2a,b, respectively. As shown in Figure 2a, the results indicate that the increased aging temperature results in a decrease of penetration values for all the modified asphalts. The linear fit analysis result found that the correlation coefficient between the penetration values and aging temperature was 0.905. Similar trends for the modified binders can be found in Figure 2b, where the penetration values are reduced when the aging time increases. The MACRMA with 20 h TFOT aging has the lowest penetration values. In addition, the penetration values have a better correlation with aging time.

The ductility indicates the plastic deformation ability and the low temperature performance of the asphalt. The test results shown in Figure 3a illustrate that the ductility values of all the asphalts reduce as the aging temperature increases from 150 to 190 °C. The lowest ductility value is 7.0 cm, which correlates to the 190 °C aged asphalt. The linear fit analysis results found that the correlation coefficient between the ductility values and aging temperature is 0.939. Additionally, the results shown in Figure 3b indicate that the ductility values reduced as the aging time increased from 0 to 20 h, and the lowest ductility value is 3.6 cm when the aging time rose to 20 h. Moreover, the ductility values also have a better correlation with aging time.

The softening point of the asphalt is the conditional temperature when asphalt reaching a certain viscosity, which indicates the thermal stability of the asphalt. The higher the softening point, the better the thermal stability of the asphalt. As shown in Figure 4a, all samples after aging have higher softening point values, and the highest softening point values reach 75.5 °C when the aging temperature is 190 °C. In Figure 4b, the samples exhibit similar trend results under various aging times. However, the softening point value is more than 76 °C when the aging time exceeds 10 h. In addition, the highest softening point value is 83.7 °C when the aging time is 20 h. Moreover, the linear fit analysis results found that the correlation coefficient between the softening point values and aging time was 0.990.

In general, it can be observed that the softening point of the MACRMA increased with the increase of aging temperature and time. However, the penetration and ductility gradually decreased. Consequently, it could be concluded that TFOT aging will increase the aging degree of leaching, regardless of aging time and temperature.

### 3.2. Viscosity

As mentioned above, the classical properties test can properly characterize the performance of crumb rubber asphalt. However, the human error of the test has large significance on its performance accuracy. Therefore, we also need to describe its performance with other indices. The viscosity is the flow characteristics of the asphalt binder to provide some assurance that it can be pumped and handled at the hot mix facility. It is also used to develop temperature–viscosity charts for estimating mixing and compaction temperature for use in mix design as a reference. Generally, a higher viscosity value results in higher mixing and compaction temperatures and may increase the energy consumption.

As shown in Figure 5a, it can be observed that the viscosity values of MACRMA increase as the aging temperature increased except at 180 °C. When the aging temperature is increased from 150 to 170 °C, the viscosity gradually increased from 3.04 Pa·s to 3.53 Pa·s. As the aging temperature rises to 180 °C, the viscosity is 3.20 Pa·s. The reason may be that the crumb rubber is degraded at high temperature, and the internal net structure is cracked into small chains, and the light component in the bitumen absorbed by the swelling action is released. However, this does not mean that the asphalt does not age. In fact, this process reflects the combined effect of the continuous swelling reaction of the rubber powder and the aging degradation on the rheology at 180 °C [33]. When the temperature continues to rise to 190 °C, the viscosity of the MACRMA increases rapidly and reaches 51 Pa·s. Therefore, it can be concluded that the production and construction temperature of microwave crumb rubber modified asphalt is recommended to be below to 190 °C.

Similarly, as shown in Figure 5b, the viscosity of MACRMA after different aging times increases rapidly from 2.93 to 4.71 Pa·s from 0 to 10 h. It might be a result of a strong swelling reaction in the modified asphalt at the initial stage of aging. It seems that the light component in the asphalt is absorbed by crumb rubbers, which cause volume expansion, and the oil content in the mixed system is reduced. Moreover, the viscosity is rapidly increased. In addition, in the 10–20 h segment, the viscosity growth tends to be stable, and reaches 5.76 Pa·s, which is the highest value of any of the modified asphalt samples. In general, the TFOT asphalts have a noticeably higher viscosity value than the unaged asphalt. This indicates that the TFOT aging causes the asphalt to stiffen at the maximum pavement service temperature, which is beneficial in improving resistance to permanent deformation.

### 3.3. G*/sin δ

The *G**/sin δ is often used to characterise the rutting resistance of an asphalt pavement at high temperature. A greater *G**/sin δ value indicates a pavement with better permanent deformation resistance. It is worth mentioning that a frequency sweep is a particularly useful test as it enables the viscoelastic properties of a sample to be determined as a function of timescale. In this study, the frequency sweep test was used to determine the *G**/sin δ values of different MACRMA samples.

The *G**/sin δ values of the MACRMA regarding an increased frequency after the aging temperature and aging time respectively are shown in Figure 6. In Figure 6a, it can be observed that with the increase of loading frequency, the *G**/sin δ values of MACRMA tend to increase gradually. All *G**/sin δ values of modified asphalt at 0.1 kPa are very close, which is not significantly different between aged and unaged asphalt. In addition, the *G**/sin δ values are higher than 1.2 kPa when the frequency increased to 1 rad/s, and some are even greater than 3.2 kPa when the aging temperature was 190 °C. At 10 rad/s, the lowest *G**/sin δ value found was 7.29 kPa when the MACRMA was unaged. However, at 150 °C or higher to, the *G**/sin δ values of aging asphalts are greater than 8.6 kPa. Among them, the MACRMA after 190 °C aging conditions have the greatest *G**/sin δ values, which reach 13.37 kPa. Generally, all modified asphalts after different aging temperature have greater *G**/sin δ values than unaged. This indicates that the aging temperature stiffens asphalts at pavement service conditions, which is beneficial to improve the resistance to permanent deformation.

Similar trends for those modified asphalts at different aging times can be found in Figure 6b. It was observed that *G**/sin δ values of all MACRMA samples tend to increase with frequency. The reason might be that the higher loading frequencies correlate well with faster driving speeds. The load action time is so short that the load dissipated quickly. Therefore, the road surface is less prone to rutting. Compared with Figure 6a, there are more significant differences in *G**/sin δ values among overall asphalt sources, and the trend is directly correlated with frequency. Moreover, when the aging time is long enough, the *G**/sin δ values of MACRMA are significantly higher than the *G**/sin δ values of MACRMA under different aging temperatures. For instance, when the aging time exceeds 10 h, the *G**/sin δ values of MACRMA is more than 16 kPa at 10 rad/s. It was concluded that the TFOT aging results in a stiffer asphalt binder, which is beneficial in improving the rutting resistance of MACRMA, regardless of the aging temperature and time.

### 3.4. Phase Angle

In Superpave specifications, phase angle is defined as the time lag between strain and stress under traffic loading and is highly dependent on the temperature and frequency of loading. It can be used as an indicator of viscosity and elasticity of binders. In general, for a perfectly elastic material, an applied load causes an immediate response, thus, the time lag is zero. Conversely, a viscous material has a relatively large time lag between load and response; its phase angle approaches 90°. Under normal pavement temperatures and traffic loadings, asphalt binders act with the characteristics of both viscous liquids and elastic solids. However, this property can be influenced by the nature of the polymer such as visco-elastic, visco-plastic, and elastic–plastic at a high temperature. In this study, all TFOT residues of MACRMA were tested at the high temperatures from 58 to 82 °C and thus exhibit viscoelastic properties.

As shown in Figure 7a, phase angles of various modified asphalts increase as the test temperature increased when the test temperature increased from 58 to 82 °C. In addition, at the frequency of 10 rad/s, the phase angles of the unaged sample reached its highest at 82 °C, and the peak value at this time was 2.54. The results indicated that the unaged sample has a greater phase angle than other modified binders and exhibited a more viscous behaviour. The phase angles of different aging samples reduced as the aging temperature constant increased compared with the unaged sample. However, no significantly different phase angles were found between 150 and 160 °C of modified asphalts. Therefore, for the asphalts tested, it could be concluded that the aging temperature has an impact on the viscoelastic characteristics of the MACRMA at high temperatures.

As shown in Figure 7b, all modified asphalts except unaged samples show increased phase angles when the test temperature is in the range of 58 to 82 °C, showing higher viscous characteristics. Compared with Figure 7a, when the aging time is long enough, the phase angles of MACRMA are significantly lower, and the growth trend is not obvious. It was concluded that, for the asphalts tested, the aging time has an influence on the elastic and viscous characteristics of MACRMA, which causes an increase in the elastic properties of modified asphalt. However, it also indicated that the phase angles are close to each other (5 and 10 h, 15 and 20 h). Therefore, it is necessary to do some further research to identify the visco-elastic behaviour of these polymer materials at high temperatures.

### 3.5. The Deformation Recovery Ability

The creep test is a simple test, in principle, with the application of a constant stress where the strain is measured accurately from the time of the application of the stress. Creep recovery is found when the stress is removed and the sample’s recovery response is monitored with time. Generally, if the material is elastic and the elasticity has not been disturbed by the creep test, it may recoil back to its original position. Otherwise, it will stay where it is.

During the creep recovery process, the elastic portion of the asphalt is gradually recovered over time, which is due to the presence of the viscoelastic properties of the asphalt and gives a certain retardation elastic property. The residual strain in the recovery stage of asphalt is defined as εP, and the initial strain in the recovery stage of asphalt is defined as εL. The ratio of εP/εL is used to characterise the deformation recovery ability of asphalt. The smaller the value of εP/εL, the greater the recovery ability of asphalt deformation [19,20].

Figure 8 shows the relationship between the εP/εL and the number of loadings for MACRMA under different aging conditions at a temperature of 60 °C and pressures of 100 and 300 Pa. It can be seen from Figure 8 that all asphalts tested in this study show an increase in εP/εL in the initial stage of loading (0~10), but with higher numbers of loads, the εP/εL values gradually becomes stable with some slight increases. The reasons might be the asphalt becomes fatigued with an increased number of loadings, which increases the irreversible deformation and deterioration ability of the deformation recovery. As shown in Figure 8a, it can be seen that, as expected, the unaged sample has the highest εP/εL value, which exhibits a higher viscosity than the other modified binders, whereas the asphalt under different temperatures of TFOT aging exhibits lower values with higher elastic properties. Additionally, similar trends for those modified asphalts under different aging times at 300 Pa can be seen in Figure 8b. However, there are no significant differences in εP/εL values between any two stresses for all samples, apart from 20 h of TFOT aging time of MACRMA.

As shown in Figure 8c,d, there are similar trends for MACRMA after different aging temperatures. At a stress of 100 Pa, there is only slight differences in the εP/εL values among 170, 180, and 190 °C of MACRMA. Additionally, no significantly different εP/εL values were found between 100 and 300 Pa. It indicated that stress has little effect on the deformation recovery ability of asphalt. Therefore, it could be concluded that TFOT aging could make the asphalt harder, which is beneficial in improving the performance of the deformation recovery of MACRMA.

### 3.6. Cumulative Strain

The cumulative strain reflects the total strain of the asphalt during the total process of loading and unloading. During the creep recovery process, this test is studied by monitoring the total stress processing over the decay load procedure.

Figure 9 shows the cumulative strain of the MACRMA under different stress conditions. As shown in Figure 9, the cumulative strain of asphalt increases with an increase in the number of stresses and loads, which is consistent with the fact that heavy traffic leads to rutting damage in roads. In addition, it can be noted that unaged asphalt has the highest cumulative strain regardless of the number of loads, while the cumulative strain of other aging samples is lower, although this value increases with the number of loads. In addition, the results shown in Figure 9a indicate that there are significant differences in the cumulative strain before and after aging conditions of the modified binders. The cumulative strain of aged MACRMA generally reduced as the aging time increased at the same number of loads under stress of 100 Pa. Similar trends for those modified binders can be found in Figure 9b, but the MACRMA exhibits a higher cumulative strain at a stress of 300 Pa compared to 100 Pa, which indicated that stress has a large effect on the resistance to the deformation of MACRMA. Additionally, there were no obvious differences among the 10, 15, and 20 h aging tests, but these values are noticeably different for the modified binders under stress of 100 Pa (Figure 9a).

Figure 9c,d show the cumulative strain of the MACRMA at aging temperatures from 150 to 190 °C at stresses of 100 Pa and 300 Pa. In Figure 9c, it was observed that the cumulative strain of aged MACRMA decreases with an increase of aging temperature at the same number of loads compared with unaged asphalt. For instance, the cumulative strain of unaged asphalt at 50 times is 29.52%, while the cumulative strain of aged asphalts is less than 20%. Thus, it seems that the anti-deformation ability of MACRMA increases. As shown in Figure 9d, the aged modified asphalts under stresses of 300 Pa show increased cumulative strain compared with 100 Pa, showing lower anti-deformation ability.

## 4. Conclusions

Based on the test results in this study, the following findings and conclusions can be observed of the properties of MACRMA before and after aging.
The performance of microwave crumb rubber modified asphalt after aging is closely related to aging time and aging temperature, both of which will increase the aging degree of asphalt, so that the penetration and elongation of asphalt decrease, while the softening point and viscosity increase.There may be a strong swelling and degradation reaction of the rubber powder during the aging process. This effect presents the opposite viscosity characteristics of the MACRMA with the aging trend. Additionally, the MACRMA with 190 °C or 20 h TFOT aging had the highest viscosity values. A lower mixing and compaction temperature (<190 °C) could be used to produce the MACRMA mixture, and thus, reducing the energy demand and retain the performance during the mixing and compaction procedures.The *G**/sin δ and phase angle values reported in this research were mainly influenced by the TFOT aging temperature and time. The aging will gradually increase the rutting factor of asphalt and reduce the phase angle, which could increase the elasticity of asphalt and improve the rutting ability at high temperatures.It can be observed that, in this study, the behaviours of creep and creep recovery of all modified asphalts were generally affected by TFOT aging. The aging effect reduces the cumulative deformation of the rubber modified asphalt, increases the elasticity of asphalt and improves its deformation resistance. Compared with the aging temperature, the aging time has a more significant effect on the deformation recovery ability of the MACRMA.

## Figures and Tables

**Figure 1 materials-12-01039-f001:**
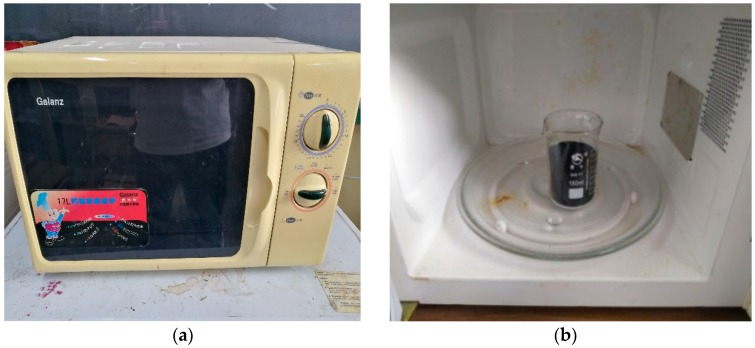
Rubber powder activation equipment: (**a**) microwave oven and (**b**) microwave-activated crumb rubber.

**Figure 2 materials-12-01039-f002:**
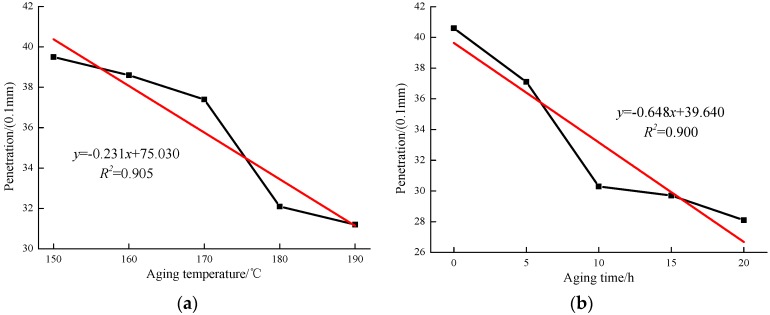
Penetration values of MACRMA at different aging conditions: (**a**) aging temperature and (**b**) aging time.

**Figure 3 materials-12-01039-f003:**
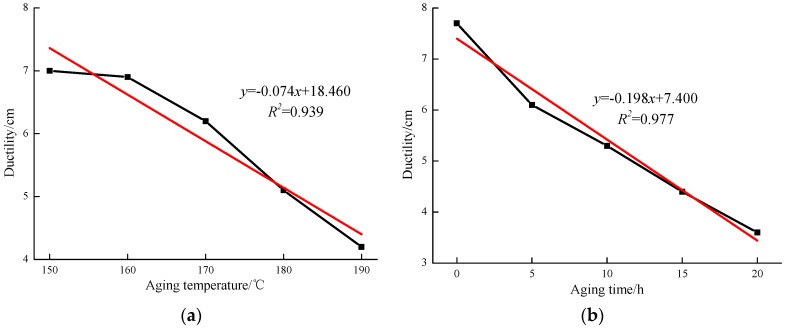
Ductility values of MACRMA at different aging conditions: (**a**) aging temperature and (**b**) aging time.

**Figure 4 materials-12-01039-f004:**
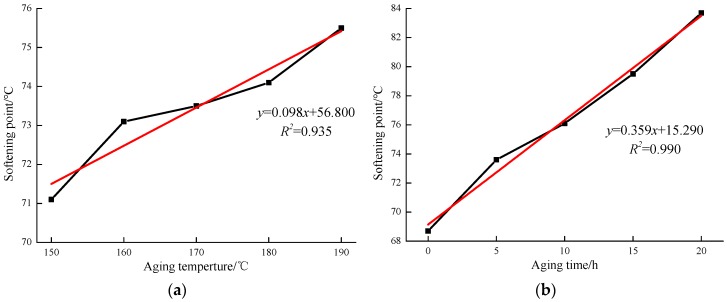
Softening point values of MACRMA at different aging conditions: (**a**) aging temperature and (**b**) aging time.

**Figure 5 materials-12-01039-f005:**
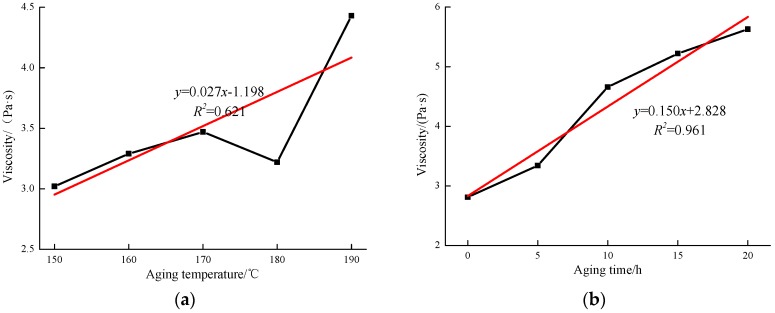
Viscosity values of MACRMA at different aging conditions: (**a**) aging temperature and (**b**) aging time.

**Figure 6 materials-12-01039-f006:**
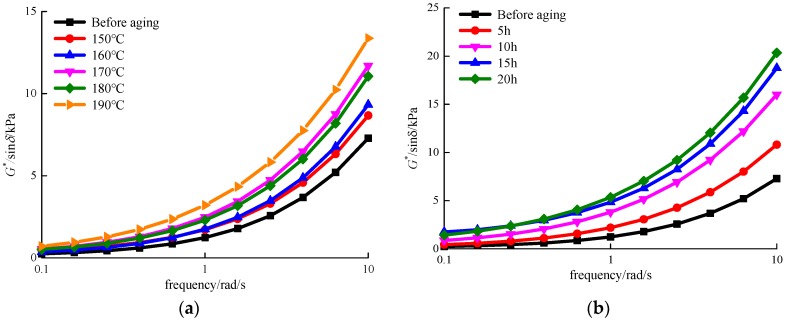
*G**/sin δ values of MACRMA at different aging condition (**a**) aging temperature and (**b**) aging time.

**Figure 7 materials-12-01039-f007:**
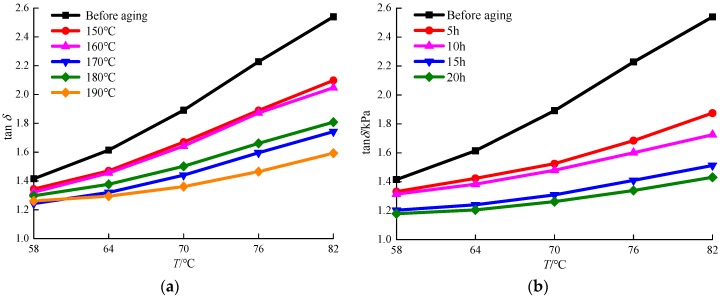
tan δ values of MACRMA at different aging conditions: (**a**) aging temperature and (**b**) aging time.

**Figure 8 materials-12-01039-f008:**
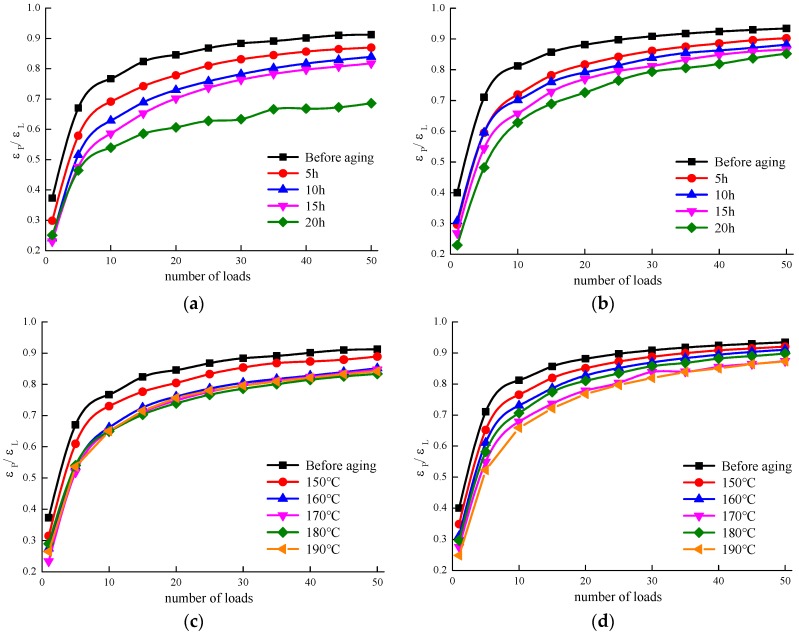
Relationship between εP/εL and load times of asphalt under different aging conditions: (**a**) different aging time at 100 Pa; (**b**) different aging time at 300 Pa; (**c**) different aging temperature at 100 Pa; and(**d**) different aging temperature at 300 Pa.

**Figure 9 materials-12-01039-f009:**
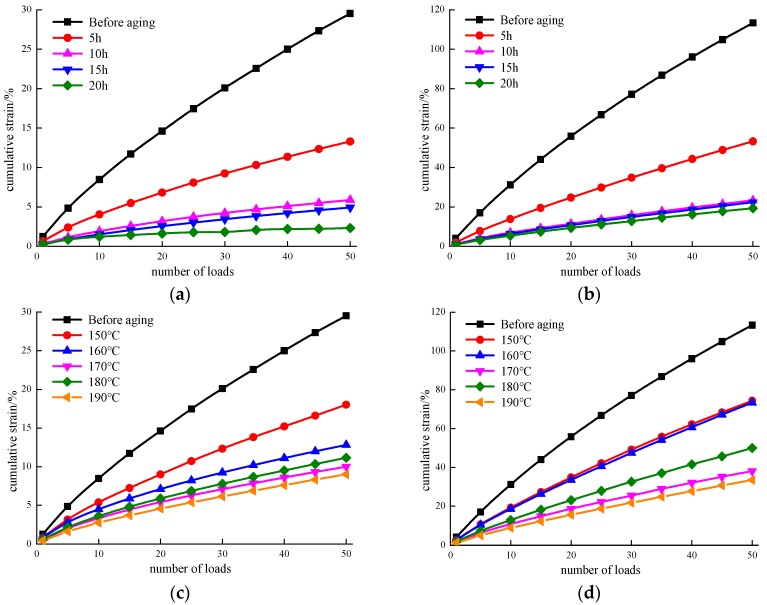
Relationship between cumulative strain of asphalt and number of loading times: (**a**) different aging time at 100 Pa; (**b**) different aging time at 300 Pa; (**c**) different aging temperature at 100 Pa; and (**d**) different aging temperature at 300 Pa.

**Table 1 materials-12-01039-t001:** Technical properties of neat petroleum asphalt.

Item		Specification	Result	Test Methods
Penetration (25 °C, 100 g, 5 s) (0.1 mm)		80~100	92	T 0604-2011 [32]
Softening point (°C)		≥42	46.2	T 0606-2011 [32]
Ductility (5 cm/min, 10 °C) (cm)		≥100	>100	T 0605-2011 [32]
RTFOT (163 °C, 85 min)	Loss (%)	≤±0.8	0.07	T 0610-2011 [32]
	Penetration ratio (%)	≥54	70	T 0604-2011 [32]
	Ductility at 10 °C (cm)	≥6	9	T 0605-2011 [32]

**Table 2 materials-12-01039-t002:** Physical and chemical specifications of rubber powder.

Item	Density (kg·m^−3^)	Water Content (%)	Metal Content (%)	Fiber Content (%)	Ash (%)	Acetone Extract (%)	Carbon (%)	Rubber Hydrocarbon (%)
Specification	260~460	<1	<0.03	<1	≤8	≤22	28	≥42
Result	302.5	0	0.009	0.065	7.3	7.2	30	52

**Table 3 materials-12-01039-t003:** Technical properties of microwave-activated crumb rubber-modified asphalt (MACRMA).

Item	Specification	Result
Penetration (25 °C, 100 g, 5 s) (0.1 mm)	30~70	40.6
Softening point (°C)	>65	68.7
Ductility (5 cm/min, 10 °C) (cm)	>5	7.7
Viscosity values (Pa·s)	2.0~5.0	2.93

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
