# Peer review of "Effect of Short-Term Aging on Asphalt Modified Using Microwave Activation Crumb Rubber"

_materials, 2019, doi:10.3390/ma12071039_

Reviewer 1 Report

On the consistency/visoscity/G*/sinδ/phase angle tests sections did you compared your results to virgin asphalt? It would be useful for the reader to have the aging of the virgin asphalt as a reference. This will also help differentiate the effects of normal oxidative aging versus the rubber enhanced aging.

On the viscosity tests line 216 you mention "While the aging temperature rise 216 to 180 °C, the viscosity is 3.20 Pa·s, Pa·s decreasing notably." can you provide an explanation of why you think this happens. Did you repeat this experiment more than once?

Did you do any cigar tube tests to tests the homogeneity of your blends? It will be useful for you to see how homogeneous your blends are over time at different sections of you cigar tube.

Author Response

We have submit  a point-by-point response to the reviewer’s comments as a PDF file.

Reviewer 2 Report

The article presents very interesting study regarding thermal ageing of asphalt modified by microwave activated crumb rubber. To the reviewer knowledge the study is original and presents new interesting test results. Nevertheless to improve the manuscript quality some additions and corrections should be implemented. All detailed remarks to the authors are given below:

1.       It is hard to call the method of modification new, as you can find publications dated in 2011 regarding this method.

2.       TFOT should be called as “thin film oven test”

3.       Table 1:

a.       Please check the “>” and “<” symbols in the requirements. For now it could be read that the tested bitumen do not comply with the requirements.

b.      Use the same format for describing units

c.       Soft point is Softening point

d.      Was ductility not tested over 100 cm?

e.      Please add all related standard specification in this table for the test. In the further part of the manuscript it makes the text lengthy and it is not required, especially when all recalled test are standardized in the whole world

4.       Table 2 – please check the descriptions not to leave single letter in the new line

5.       Please add the table with properties of ready asphalt modified by crumb rubber. From the further parts of the manuscript it seems that authors have all related data

6.       Why did authors not compared achieved results to non-modified asphalt?

7.       P.2. line 115 – was the temperature to heat the asphalt not controlled?

8.       P.2. line 124 and further – Part about basic asphalt tests is lengthy and could be removed

9.       P.3 line 153 – please check if AASHTO MP19-1 have correct name and number and describes RCRT test. From author knowledge and search it describes MSCR test, which methodology is a bit different

10.   P3. Line 162 – it is a repetition of the penetration grade test and could be removed

11.   Fig 2 and further – the reviewer is not sure if for such small amount of points R2 indicator is good. Similar correlation coefficient could be achieved even for different function, not only linear. And even small differences in test results tendencies (fig5a) impacts R2 indicator strongly

12.   Fig 2 – use the same description type of units in the whole manuscript

13.   P. 6 line 187-189 – some mechanical errors with “enters” were made. Please correct

14.   P. 6 line 192 – does in this case R&B temperature really indicates better thermal stability and not the ageing of the material? Going further with this interpretation it could be “better” for asphalt or mixture to age it for more time to achieve better thermal stability

15.   P.6 line 205 – what does authors mean speaking about “human error”? The result for 180°C in figure 5a? Why authors did not repeat the test for this temperature?

16.   3.2 Viscosity – can author add any comment regarding technological values of viscosity (0.2 and 2 Pa.s) in comparison to achieved test results?

17.   P7. 232 – the description of the common test is lengthy and can be shortened

18.   Fig 6 – add the test temperature for the frequency sweep test.

19.   P 8 line 280 – tan d or phase angle have the value of 2.54? please provide correct description

20.   P. 8 line 294 – polymer material? Or crumb rubber modified?

21.   P. 8 line 295 – please provide more descriptive name of the paragraph

22.   Conclusions – check the text style

23.   References – there are strange “[J]” symbol in each document. Please check the style. And check the typos (for example - reference 19 – “Resrarch”)

Author Response

We have submit  a point-by-point response to the reviewer’s comments as a PDF file.

Round  2

Reviewer 1 Report

Authors have addressed all my comments.